# Readout and delayed transmission of initial afferent V1 activity in decisions about stimulus contrast

**Kieran S Mohr\*, Simon P Kelly**

Cognitive Neural Systems Lab, School of Electrical and Electronic Engineering and UCD Centre for Biomedical Engineering, University College Dublin, Dublin, Ireland

## eLife Assessment

This **important** study reports that EEG recordings of the earliest stage of information processing in human visual cortex can be used to predict subsequent choice responses. The findings provide novel, **convincing** evidence for integrative processing in low-level sensory cortices at the level of scalp-recorded potentials, with the exact nature of the neural signals at the single cell level to be determined. The paper is likely to be of interest to neuroscientists interested in the contribution of early sensory signals to decision making.

**\*For correspondence:**
kiemohr@gmail.com

**Competing interest:** The authors declare that no competing interests exist.

**Abstract** Initial afferent activation of V1, indexed by the C1 component of the human VEP, is often considered to be a rudimentary stage of visual processing, operating mostly as a conduit for later stages with limited cognitive penetrability. The full suite of visual analysis entails activity across several visual areas and feedback from later areas to earlier ones. This raises the question of whether the early sensory representation indexed by the C1 is read out for perceptual decisions or whether it is passed over in favour of more advanced representations. To address this question, we asked whether the C1 would predict time-pressured stimulus contrast comparisons independently of physical stimulus conditions, a phenomenon known as choice probability. We found that the C1 did this for a narrow range of response times, indicative of decision readout since the C1 is a transient signal. This effect could not be accounted for by stimulus differences, choice history, or any other choice-predictive signal that we could identify in either the time or frequency domain, either before or after target onset. It also preceded the onset of evidence-dependent decision formation estimated from the centroparietal positivity by tens of milliseconds, together providing an approximate timeline of early evidence readout and its delayed impact on the decision.

## Introduction

V1 is an important hub in the visual system, being the main conduit for geniculo-cortical afferents en route to a vast hierarchical structure of functionally specialised cortical areas (**Felleman and van Essen, 1991**; **Nassi and Callaway, 2009**; **van Essen and Maunsell, 1983**; **Zeki, 1992**), which in turn send extensive feedback connections back to V1 (**Kravitz et al., 2013**; **Markov et al., 2014**; **Muckli and Petro, 2013**). For this reason, it has been extensively studied across many species through multiple neural activity assays, which include the C1 component (70–90 ms) of the visual evoked potential (VEP) in human electroencephalography (EEG; **Clark et al., 1994**; **Di Russo et al., 2002**; **Jeffreys and Axford, 1972**; **Mohr et al., 2024**). Because of V1's rich connections, its small receptive field sizes and its selectivity for basic stimulus features (**Felleman and van Essen, 1991**; **Hubel and Wiesel, 1962**; **van Essen and Maunsell, 1983**), its role has been likened to that of a 'blackboard' where crude

information is initially written by afferents but then passed up to higher visual areas, which in turn rewrite the blackboard via feedback modulations (*Bullier, 2001*; *Roelfsema and de Lange, 2016*). Indeed, *Leopold, 2012* argued that V1 likely serves mostly as a conduit to these later areas rather than being an important site for generating perception in itself. However, while it is clear that feedback operations are necessary for a variety of complex visual tasks involving scene segmentation (*Roelfsema and de Lange, 2016*), basic tasks such as discriminating simple stimulus features or contrast gradients could in principle be informed by any stage of visual analysis, including afferent V1 activity where the visual scene is first represented and at high spatial resolution (*Zeki, 1978*). Nevertheless, it is possible that such early representations are overlooked by decision readout regardless of their task utility if they are part of an automatic process that is not involved in generating perception (*Hochstein and Ahissar, 2002*). Indeed, many visual features are represented by multiple visual areas and could facilitate such an early-automatic versus late-perceptive dichotomy, including orientation (*Felleman and van Essen, 1987*; *Hubel and Wiesel, 1969*), motion direction (*Albright, 1984*; *Felleman and van Essen, 1987*; *Geesaman and Andersen, 1996*), stereoscopic disparity (*Neri et al., 2004*; *Uka et al., 2005*), and contrast (*Sclar et al., 1990*). Thus, the question of whether perceptual decisions make use of the earliest available sensory representations or wait for later ones remains open.

The question of whether a given sensory neural representation is used for a perceptual judgment has classically been studied through the construct of choice probability, broadly defined as a correlation across trials between a sensory neural signal and the behavioural choice, independent of the physical stimulus (*Britten et al., 1996*). Choice probabilities have typically been examined at the level of single-neuron firing rates and have been observed in many visual areas and for many tasks, appearing selectively in task-relevant neurons, suggestive of choice readout (*Bondy et al., 2018*; *Britten et al., 1996*; *Britten et al., 1996*; *Cohen and Newsome, 2008*; *de Lafuente and Romo, 2006*; *Dodd et al., 2001*; *Jasper et al., 2019*; *Kang and Maunsell, 2020*; *Liu and Newsome, 2005*; *Nienborg et al., 2012*; *Nienborg and Cumming, 2006*; *Nienborg and Cumming, 2009*; *Parker and Newsome, 1998*; *Price and Born, 2010*; *Romo et al., 2002*; *Uka and DeAngelis, 2004*; *Wimmer et al., 2015*; *Yu and Gu, 2018*). They tend to grow over time from evidence onset and are stronger in hierarchically higher than lower visual areas (*de Lafuente and Romo, 2006*; *Jasper et al., 2019*; *Krishna et al., 2021*; *Nienborg et al., 2012*; *Nienborg and Cumming, 2006*), suggestive of choices that are largely driven by late stages of visual analysis. Indeed, optogenetic silencing of secondary visual areas suggested that these higher areas were required to perform a contrast change detection task (*Goldbach et al., 2021*). Nevertheless, *Resulaj et al., 2018* demonstrated that the first one or two V1 spikes in the first 40–80 ms of its response in an orientation discrimination task were necessary for successful task performance in mice, demonstrating the crucial role of early visual activity. Moreover, the tendency for choice probability to grow over time may be driven by decision-related feedback from higher to lower areas rather than increasing choice weighting, as demonstrated by an early-peaking psychophysical kernel alongside late-peaking choice probability (*Nienborg and Cumming, 2009*). In fact, in studies that performed a time-resolved analysis, onset latencies of choice probability could be as early as 50–100 ms (*Cohen and Newsome, 2008*; *Cook and Maunsell, 2002*; *Price and Born, 2010*; *Uka and DeAngelis, 2004*), suggesting uptake of early evidence, and this sometimes coincided with the initial transient response of sensory neurons (*Dodd et al., 2001*; *Uka and DeAngelis, 2004*), although not always (*Kang and Maunsell, 2020*; *Liu and Newsome, 2005*; *Nienborg and Cumming, 2006*). However, in the critical case of initial afferent V1 activity, choice probability reports have been mixed, with some finding no or weak choice probability (*Goris et al., 2017*; *Jasper et al., 2019*; *Lange et al., 2023*; *Niemeyer et al., 2022*; *Nienborg and Cumming, 2006*; *Ziemba et al., 2024*), some finding it, but late in the response (*Krishna et al., 2021*; *Ziemba et al., 2024*) or without a time-resolved analysis (*Bondy et al., 2018*; *Boundy-Singer et al., 2025*; *Nienborg and Cumming, 2014*), and some finding it shortly after or immediately upon the onset of the V1 response (*Cone et al., 2024*; *Kang and Maunsell, 2020*; *Palmer et al., 2007*). This inconsistency in V1 choice probability is perplexing since many studies reporting null findings involve tasks for which V1 neurons are well suited, with good neurometric performance, and indeed provide the input to higher visual areas where more consistent choice probability results are found (*Felleman and van Essen, 1991*). This suggests that the principles underlying decision readout are not fully determined by neurometric performance and hints that one such principle may be to favour higher-level sensory representations for decision readout.

While the question of readout of initial afferent V1 activity remains unclear in the animal literature, even less is known about it in the human literature where no study, to our knowledge, has directly probed the role of initial afferent V1 activity in choice formation. Although several demonstrations of choice-related variations in the steady-state VEP (SSVEP) have been shown (*Campbell and Kulikowski, 1972*; *Grogan et al., 2023*; *O'Connell et al., 2012*), the SSVEP cannot distinguish bottom-up afferents from recurrent or top-down influences on sensory responses. Some insights into the onset latency of sensory readout in humans have stemmed from analyses of the centro-parietal positivity (CPP), a signal that has been linked to decision formation (*Kelly and O'Connell, 2013*; *O'Connell et al., 2012*). The CPP is believed to index the accumulation of sensory evidence towards a decision threshold because it starts to build up following the onset of decision-related sensory evidence, its buildup rate scales with the strength of sensory evidence, it peaks at a time aligned to response with an amplitude consistent with the reaching of a threshold at the downstream motor level, and it is observed regardless of the need for an immediate motor response (*Kelly and O'Connell, 2013*; *O'Connell et al., 2018*; *Twomey et al., 2016*). CPP onsets can range from around 150–300 ms across tasks (*van den Brink et al., 2021*; *Geuzebroek et al., 2023*; *Kelly and O'Connell, 2013*; *Loughnane et al., 2016*; *McCone et al., 2025*; *Newman et al., 2017*; *Steinemann et al., 2018*), and this may even underestimate the onset of true evidence readout since CPP onset can sometimes precede evidence-driven buildup (*Devine et al., 2019*). At 150 ms, advanced sensory representations are available that can, for example, distinguish between scene categories (*Fabre-Thorpe, 2011*; *Hegdé, 2008*). Therefore, such a long delay in CPP onset may imply a reliance on late sensory evidence considering that widespread cortical activation is evident in the first 100ms (*Foxe and Simpson, 2002*). However, although this seems suggestive of decision readout that passes over the earliest available sensory evidence from initial afferent V1 activity, it remains possible that such early evidence is used but that there is a transmission delay between sensory and decision-related areas.

Here, we pursued a direct investigation into whether initial afferent V1 activity is read out in human choice formation by testing for a choice probability effect in the C1 component (70–90 ms) of the VEP. The C1 is thought to predominantly represent initial afferent V1 activity due to its latency as the earliest VEP component and the unique consistency with V1 morphology of its topographic shifts across retinotopic space (*Clark et al., 1994*; *Di Russo et al., 2002*; *Jeffreys and Axford, 1972*; *Kelly et al., 2013*), and, accordingly, V1's superior quantitative fit to the C1 compared to other visual areas (*Mohr et al., 2024*). Because the C1 is highly sensitive to contrast (*Gebodh et al., 2017*), we used a contrast-based task, which is also ideal to probe questions of early versus late decision readout because contrast is represented throughout the visual system (*Sclar et al., 1990*). To take the extra step of inferring decision readout of the C1 from its choice probability, we made a number of design and analysis choices. Firstly, to emphasise the importance and hence potential readout of early V1 activity, we used tight response deadlines. Reasoning that the readout of an early transient signal such as the C1 would impact a delimited range of speeded response times, before which its information was not yet available and after which its contribution would be diluted by later visual cortical activation (see *Figure 1—figure supplement 1* for a motivating simulation), we set response deadlines of 400 ms and 600 ms in separate blocks to together elicit a wide enough range of speeded RTs to test for this. This RT contingency also helped to rule out some interpretations of choice probability that are not predicated on decision readout, such as mediation by attention, which would have no reason to predict such an intermediate RT range (*Figure 1—figure supplement 1*). Secondly, capitalising on the global viewpoint of EEG, we could further rule out a mediation-based relationship by comprehensively measuring all detectable choice-predictive signals and controlling for them in C1-choice regression analyses. By demonstrating that the C1 met all these criteria, we provide evidence for early readout of sensory evidence in the context of speeded contrast comparisons. By comparing the latency of this evidence readout (approximately 80 ms) with the onset of evidence-dependent CPP buildup (approximately 145 ms) and the fastest above-chance RTs (approximately 200 ms), we also provide estimates for transmission delays between sensory, decision and motor stages of choice formation in the order of tens of milliseconds.

## Results

Participants (N=18) compared the contrast of two arrays of gratings presented simultaneously in the upper left and lower right quadrants of the visual field while EEG was recorded (*Figure 1A*), indicating

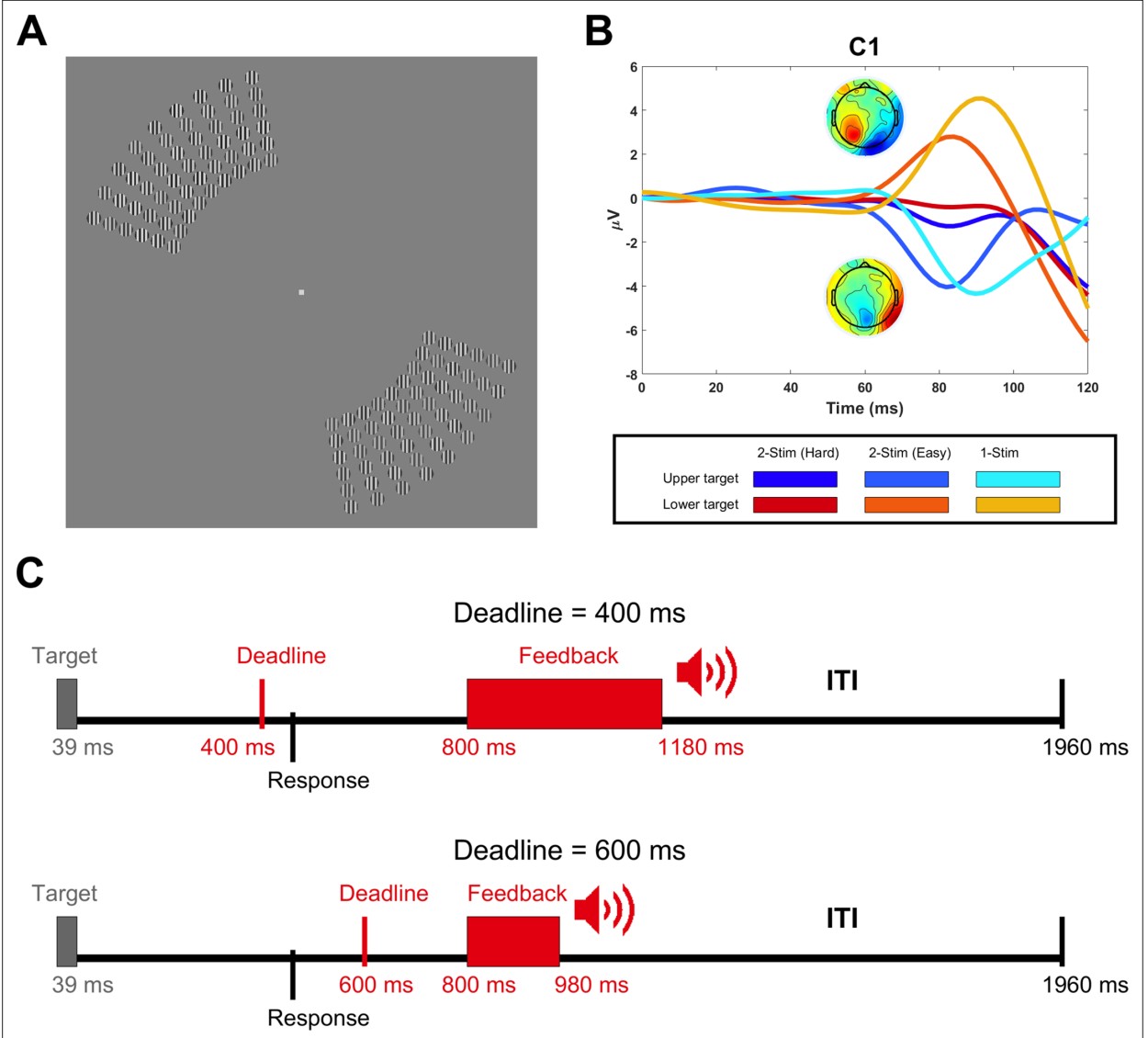

**Figure 1.** Stimulus layout and experimental design. (**A**) Stimulus layout showing the upper-left and lower-right arrays of gratings (showing 6 x 9 arrays rather than the full 9 x 9 arrays for better visibility). Participants indicated which array had higher contrast by pressing the up or down arrow key with their left or right index finger, with the response mapping counterbalanced across blocks. (**B**) Grand average waveforms in C1 electrodes (selected for each participant individually based on maximal C1 differences between upper- and lower-field arrays from the easy block of trials) for upper targets (blue) and lower targets (red) in the main task blocks (darkest shade), the easy blocks (intermediate shade) and the blocks in which only the target array appeared (lightest shade). The topographies show the C1 (80–90 ms) for upper and lower-field targets in the easy condition. (**C**) Timeline of two example trials, one with the 400 ms deadline and a late response (top) and the other with the 600 ms deadline and a timely response (bottom). The feedback only told participants whether or not they were on time. Accuracy feedback was given summarily at the end of each block.

The online version of this article includes the following figure supplement(s) for figure 1:

**Figure supplement 1.** Results of the simulation designed to illustrate the expected dependency of C1 choice probability on response time.

their choice by pressing an up or down arrow key with their left or right index finger, with this mapping counterbalanced to orthogonalise perceptual space and motor laterality (see Methods). The task was rendered difficult by individually titrating to 55% accuracy, and this balance between correct and erroneous responses was important to facilitate well-powered choice analyses that could control for physical stimulus differences. Further, a tight deadline of either 400 or 600 ms was imposed to induce an appropriate range of speeded RTs. The overall aim was to test whether sensory evidence contained in the C1 could be read out for decisions about stimulus contrast, especially impacting an intermediate range of RT. This analysis relied on C1 amplitude being diagnostic of contrast differences between the

upper and lower arrays, which was achieved by the stimulus layout (*Figure 1A–B*) because C1 polarity reverses between the upper and lower visual field (*Jeffreys and Axford, 1972*) and its amplitude varies by contrast (*Gebodh et al., 2017*).

## Choice-predictive signals

It was important to rule out a C1-choice relationship that was mediated by other choice-predictive variables. For example, a pre-existing, random attentional bias toward one side could enhance both the C1 (*Qin et al., 2022*) and the perceived contrast associated with that side (*Liu et al., 2009*) without a direct causal link between the latter two. To control for this, choice regressions should include as many choice-predictive covariates alongside the C1 as possible, including lateralised spectral indices of attentional allocation between the target locations, facilitated by their positions in the left and right visual hemifields (*Foxe and Snyder, 2011*). We therefore first carried out a comprehensive search for choice-predictive signals across the EEG activity. We did this in both the time domain and time-frequency domain by visual inspection of the difference across choice outcomes in EEG activity from 600 ms before target onset to 400 ms afterwards (see Methods). Our goal here was to be liberal to ensure we did not miss any choice-predictive signals and because including false positives as covariates would not adversely affect the analysis. In the time domain, we found seven signals that exhibited a larger amplitude for upper-field choices (*Figure 2A*): a pre-target midline positivity, a dipolar fronto-occipital deflection at approximately 150 ms, and a sequence of bilateral occipital deflections at 200 ms and 260 ms. In the time-frequency domain, we found six signals (*Figure 2B*): a pre-target contralateral decrease in alpha and beta activity (cluster 1), a post-target contralateral decrease in a wide band from theta through to beta (cluster 2), a pre-target decrease in ipsilateral delta activity (cluster 3), a pre-target decrease in frontal beta activity (cluster 4), a post-target increase in ipsilateral theta activity (cluster 5), and a post-target decrease in ipsilateral delta activity (cluster 6). Contralateral suppression ('desynchronisation') of the alpha band was particularly strong and long-lasting and was consistent with spatial attention aligning with the to-be-chosen location (*van Dijk et al., 2008*; *Foxe and Snyder, 2011*; *Kelly et al., 2006*; *Klimesch, 2012*; *Thut et al., 2006*).

## Behavioural results

In line with our intention that task difficulty would be high while keeping participants engaged throughout the experiment, a mixed-effects logistic regression model indicated that task performance remained steady at approximately 55% accuracy with a marginal drop of 0.25% per block during the first EEG session ($t(18192)=-1.98$, $p<0.05$), no significant change across the second session ($t(18110)=-1.42$, $p>0.1$), and no significant difference between sessions ($t(36321)=0.79$, $p>0.1$). Accuracy was higher under the 600 msec deadline (57.6%) than the 400 msec deadline (53%; $t(36321)=8.79$, $p<0.0001$; *Figure 3B*), while response times (RTs) were slower (345 ms vs 267 ms; $t(36100)=83.7$, $p<0.0001$) and more variable (paired-samples $t(17)=2.48$, $p<0.05$) under the 600 ms deadline, reflecting a typical speed-accuracy trade-off. Error responses were faster (298 ms) than correct responses (313 ms; $t(36100)=14.8$, $p<0.0001$), but no less variable ($t(17)=0.87$, $p>0.1$). There was an overall tendency to favour the lower-field array, which was chosen in 52% of trials ($t(36322)=10.1$, $p<0.0001$), and a significant choice history effect ($t(36321)=27.03$, $p<0.0001$) driven by a tendency to switch responses from one trial to the next, which resulted in lower-field choices on 58.7% of trials following a trial with an upper-field choice. This bias varied by a quadratic function of RT ($t(24611)=4.89$, $p<0.0001$) and predominated in the fastest RTs (*Figure 3—figure supplement 2*).

Accuracy also varied significantly by a quadratic function of RT ($t(36099)=3.9$, $p<0.0001$), starting at chance level for the fastest responses, peaking at intermediate responses, and falling again at slow responses (see *Figure 3C*). This pattern was qualitatively similar for both deadline conditions (*Figure 3D*), though, after controlling for the quadratic effect of RT on accuracy, a main effect of deadline remained ($t(36098)=2.8$, $p<0.01$), indicating that the longer deadline had an accuracy advantage over the shorter deadline above and beyond that attributed to their differing RT distributions. Error shading indicates standard error of the mean derived from mixed effects regression models with 1817 observations taken from N=18 participants.

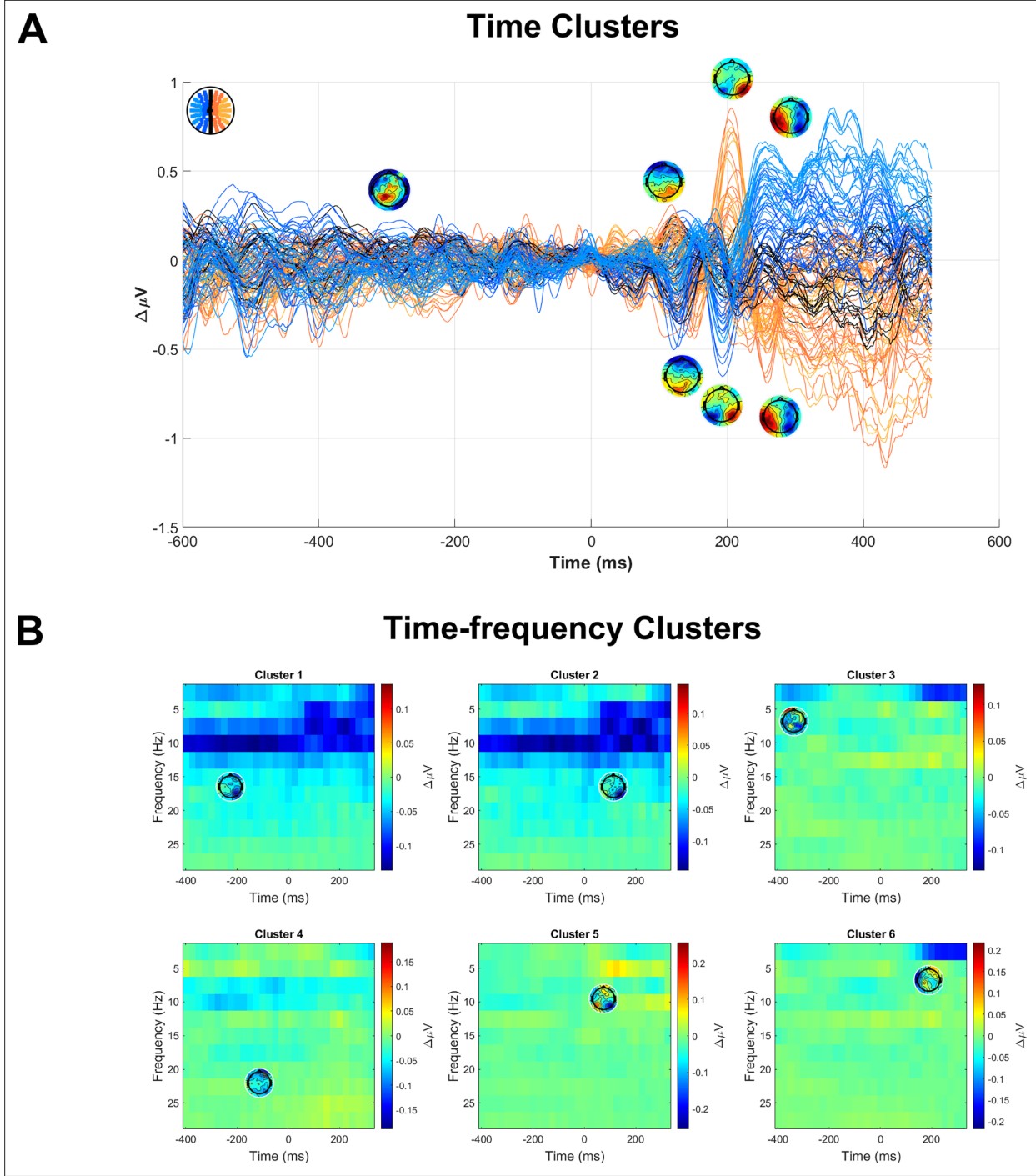

**Figure 2.** Choice-predictive signals used as covariates to control for in C1-choice regression models. (**A**) Butterfly plot showing the subtraction of lower-chosen trials from upper-chosen trials, including topographies indicating the identified signals. In cases where the topography included both positive and negative poles, electrodes were chosen from both poles as separate signals. (**B**) Time-frequency plots of the same subtraction applied after converting to the time-frequency domain, showing topographies of the six choice-predictive signals identified.

### C1 choice probability at intermediate response times

To test for C1 choice probability while controlling for other choice-predictive variables, we used a mixed-effects logistic regression model with random intercepts to predict upper- vs lower-target choices as a function of C1 amplitude and a range of covariates. These covariates included the set of choice-predictive signals identified earlier, along with target location and the previous trial's choice.

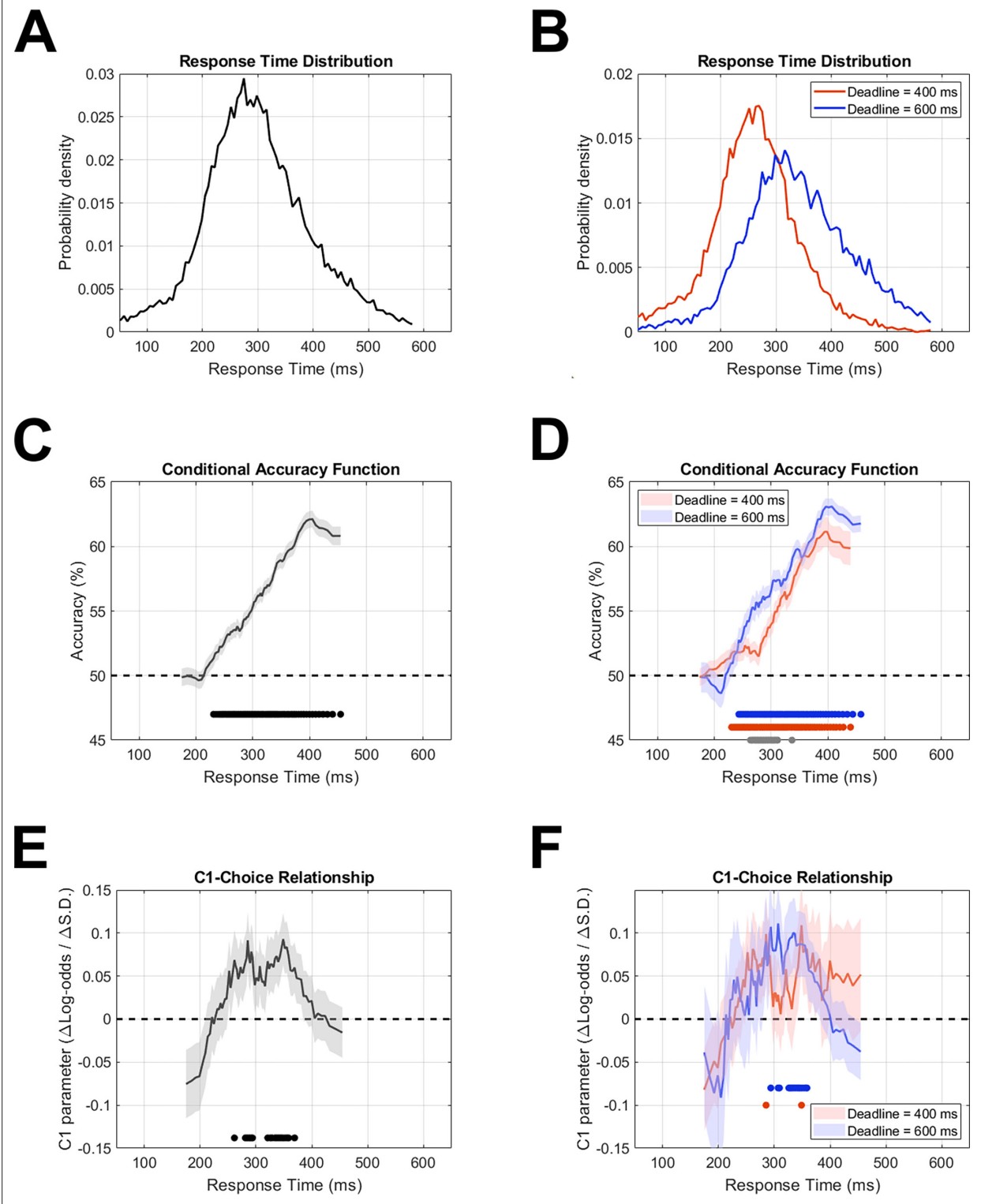

**Figure 3.** Accuracy and C1-choice relationship as a function of response time. The sliding RT windows encompassed a range of 20 percentile points each, centred at each unit percentile from 10% to 90%. (**A–B**) Response time distributions for all trials (**A**) and for each deadline condition (**B**). (**C–D**) Accuracy (percent correct) across the same RT windows. Black circles at the bottom of panel C show RT windows where accuracy was significantly above chance level. Red and blue circles at the bottom of panel D show the same for the 400 ms and 600 ms deadline respectively, while grey circles show windows where there was a significant accuracy difference between them. (**E–F**) C1 coefficients in logistic models of choice, for the same series of RT windows, multiplied by –1 for the figure so that positive values would correspond to choice-congruent C1 amplitude. Circles at the bottom indicate

*Figure 3 continued on next page*

*Figure 3 continued*

significant windows in the same way as in panels **C**–**D**. See *Figure 3—figure supplement 1* for a regression performed with C1 as the dependent variable, comparing effects of current and previous choices. Error shading in panels C-F indicates standard error of the mean derived from mixed effects regression models with between 1390-7268 observations (depending on the combination of RT window and deadline condition) taken from N=18 participants.

The online version of this article includes the following figure supplement(s) for figure 3:

**Figure supplement 1.** Models predicting C1 amplitude as a function of choices on the current trial (**A**) and the previous trial (**B**).

**Figure supplement 2.** Percentage of trials in which the upper target was chosen as a function of choices on the previous trial.

The inclusion of choice history was motivated by studies showing that prior choices can influence both choices and sensory responses in upcoming trials (*St John-Saaltink et al., 2016*; *Macke and Nienborg, 2019*), which means that choice history could mediate a choice probability effect in the same way as target location.

Because of our expectation of an RT-dependent C1 choice probability effect, we included interactions with a quadratic function of RT for all predictors. This resulted in three regressors for each predictor (its main effect as well as a linear and quadratic RT interaction term). Thus, because the model included the C1, target location, choice history, seven time domain signals and six time-frequency domain signals, it had 48 regressors in total, as well as the fixed and random intercepts. The quadratic RT interaction with C1 was significant ($t(26393)=4.96$, $p<0.0001$) with beta coefficients for the main effect, linear term, and quadratic term of 0.56, –3.68, and 5.32, respectively, indicating that C1 was choice predictive for an intermediate range of RTs even when accounting for the potential mediation by the other choice predictive signals. To ensure this analysis was robust against multicollinearity, we repeated it with two regressors omitted such that the maximum correlation coefficient magnitude amongst the remaining ones was 0.5, which did not alter the results. Importantly, the direction of the effect (more negative C1 for upper-array choices) was congruent with the direction of stimulus-driven C1 amplitude changes.

The C1-RT interaction coefficients estimate peak C1 choice probability at 346 ms. To visualise this RT dependency, we applied a sliding RT window and carried out the same model at each window without the RT interactions (*Figure 3E*). To estimate the onset of the RT range and compare it with the onset of above-chance accuracy, we repeated this sliding RT procedure in 10,000 bootstrap samples (see Methods). This indicated that above-chance accuracy emerged at RTs of 199 ms (95% CI [172,227] ms) and that C1 choice probability emerged at RTs of 232 ms (95% CI [186,304] ms), with the difference between them not reaching significance (95% CI [–6,99] ms). The offset RT for C1 choice probability was estimated at 401 ms (95% CI [298,460] ms).

Finally, although controlling for covariates rules out the possibility that they fully account for the C1-choice relationship, it is possible that they could account for it partially, which would manifest as a reduction in the magnitude of the C1 coefficient in the model. To check for this, we compared models with and without the choice-predictive signals outlined above, focusing on the identified RT range of 230–400 ms so that the RT interaction could be removed, allowing for a simple comparison of one coefficient. The C1 coefficient was, in fact, larger with the covariates included (0.058) than without (0.04), suggesting that rather than explain the C1-choice relationship, their accounting in the regression instead allowed the C1-choice relationship to be seen more clearly. The model nevertheless remained significant without the additional regressors ($t(20203)=2.7$, $p<0.01$).

## Choice history modulates C1 amplitude

The above analysis demonstrated a link between the C1 and choices that was not explained by any other choice-predictive variable, suggesting a direct cognitive interface at the level of initial afferent V1 activity. To probe a further possible interaction between cognition and the C1, we investigated choice history bias, which has been found to have an impact on other sensory signals (*St John-Saaltink et al., 2016*; *Macke and Nienborg, 2019*). To do this, we repeated the initial choice model except that we swapped the choice and C1 variables so that C1 was the dependent variable, allowing both current and previous choices to be entered together as regressors. This revealed significant quadratic RT interactions for both current trial choices ($t(24611)=3.74$, $p<0.001$) and previous trial choices ($t(32072)=2.72$, $p<0.01$), indicating that both were predictive of C1 amplitudes for a restricted range

of RTs and could not account for one another. We repeated this model without current trial choices, and the choice history bias was almost identical. Both effects were congruent in the sense that the direction of the effects was consistent with the effect of physical target location, which means that this choice history bias operated in the opposite direction to the behavioural choice history bias, which was a tendency for response switching. We again visualised the RT dependence of the effects by applying a sliding RT window (*Figure 3—figure supplement 1*). Unsurprisingly, given the equivalent architecture of this model to the original, the effect of current-trial choices on the C1 was almost identical to before (compare *Figure 3—figure supplements 1A and 3E*). Remarkably, however, the effect of the previous trial's choice on the C1 followed a very similar profile of RT dependence (*Figure 3—figure supplement 1B*). As with the direction of the bias, this RT contingency was also different to that of the behavioural bias, which was dominated by fast RTs (compare *Figure 3—figure supplements 1 and 2*).

## The onset of evidence-dependent CPP buildup implies a delay to the accumulator

While C1 choice probability was the primary focus of this study, it was of further interest to place this effect within the larger timeline of choice formation. Therefore, we next sought to estimate the delay at which sensory information enters the deliberative decision process. To do this, we sought to estimate the onset of sensory evidence accumulation via the centro-parietal positivity (CPP), which has been closely tied to decision formation (*Kelly and O'Connell, 2013*; *O'Connell et al., 2012*). Because evidence accumulation can sometimes begin prematurely and accumulate noise (*Devine et al., 2019*), the onset of CPP buildup is not a reliable marker of when real sensory evidence enters the accumulator, so we instead focused on when the slope of the CPP began to differ between easy and hard blocks to determine when it first became evidence-dependent. We did this using response-locked epochs because transient potentials evoked by sudden stimulus onset in the target-locked epoch obscure the onset of the CPP, whereas they are reduced by temporal jitter in the response-locked average. We used mean RT to convert these latency estimates to a delay from stimulus onset.

The CPP for easy and hard blocks, as well as blocks where only one of the two stimulus arrays was presented, is shown in *Figure 4A–B* for fast and slow RTs, respectively. To capture uncertainty in the estimate of when the CPP slope began to differ between easy and hard blocks, we carried out 10,000 bootstrap samples, and in each one, we applied a sliding 50 ms window in 10 ms intervals to determine the first window where CPP slope in the easy block was significantly higher than in the hard blocks, remaining so up to its peak. This estimated a median delay from stimulus onset of 159 ms (95% CI [59,199]). To corroborate this estimate, we repeated the procedure for RTs above and below the median to produce separate converging estimates. This yielded a median of 141 ms (95% CI [131,221]) for fast RTs and 148 ms (95% CI [118,238]) for slow RTs. The average of all three yields an estimate of 149 ms. Comparing this with the latency of the C1 component (80 ms) suggests a delay for this sensory information to enter the accumulator of approximately 70 ms.

## Other VEP components

To our knowledge, this is the first study to implement a task involving contrast comparisons across space while measuring EEG. As such, we present the full target-locked waveform and its difference by target location and choice in *Figure 5* to take a broader view beyond the C1. The basic waveform (*Figure 5A*) included many typical VEP components including the C1, P1 (120 ms), N1 (130 ms), N2 (180 ms), and P3b/CPP (240 ms). The target-related difference waveform (*Figure 5B*) consisted of C1 (80 ms) and C2 (110 ms) components, followed by a sequence of alternating bilateral components at 150 ms, 200 ms, and 260 ms. A similar sequence of components was present in the choice-related difference waveform (*Figure 5C*) except that the C1 component was not clearly identifiable, likely due to the dependency of its choice relationship on RT. Both the target and choice difference waveforms also included an apparent deflection at approximately 40 ms; although there have been reports of an 'N40' component for foveally presented stimuli (*Carretié et al., 2024*; *Proverbio et al., 2021*), the component shown here is unlikely to be a stimulus-evoked activation because its topography pattern is also shared by similarly sized components at 0 ms and –40 ms. It is therefore more likely to be a pre-stimulus oscillation, possibly related to the choice history bias.

Finally, since choice probability studies in the animal literature typically show that choice probability grows over the course of a trial, we generated a comparable metric by using linear discriminant

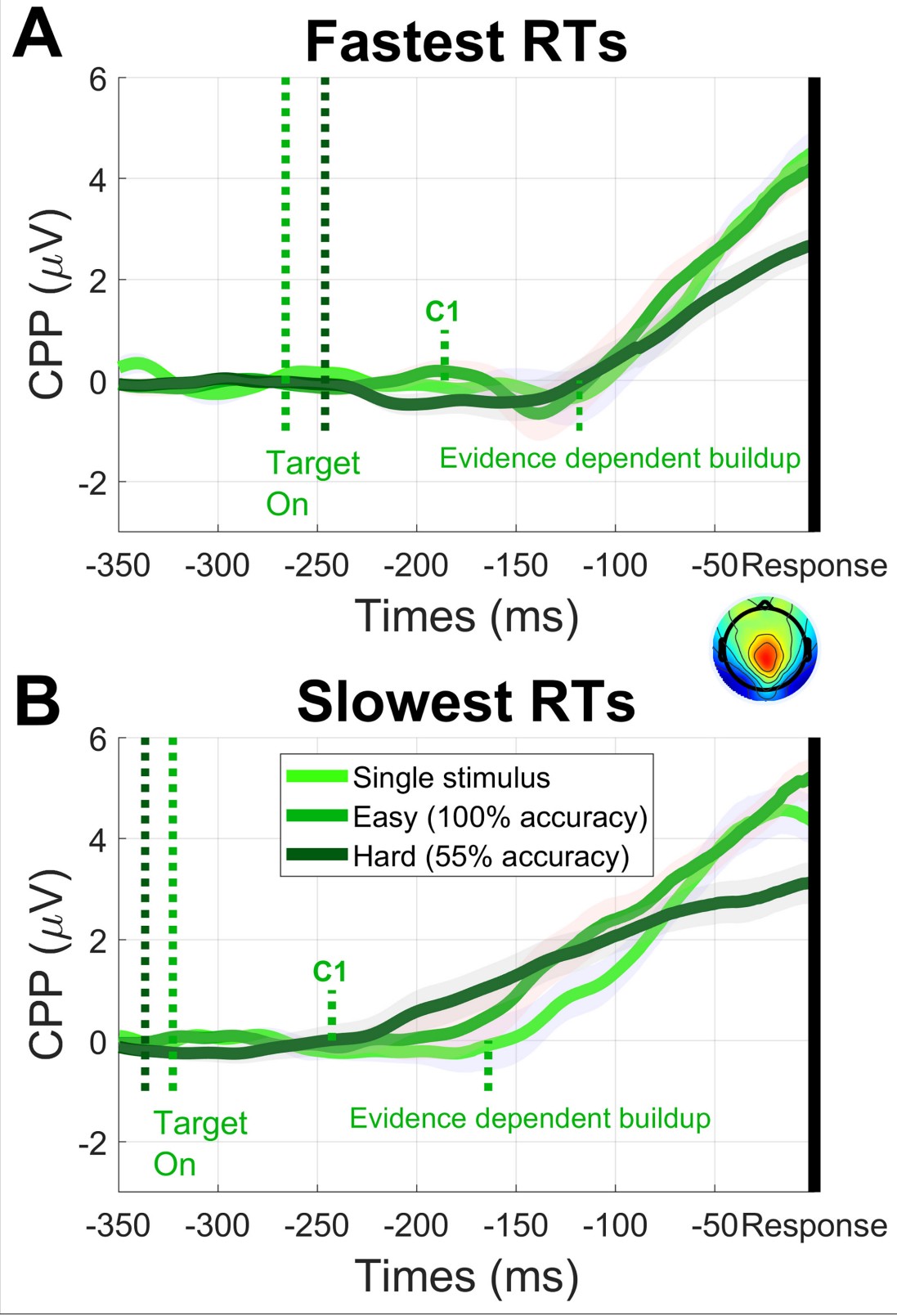

**Figure 4.** Response locked CPP in the 600 ms deadline condition for blocks of hard trials with 55% accuracy (dark green), easy trials with ceiling performance (mid-green) and trials in which only one of the two stimulus arrays appeared (light green), shown for RTs faster (**A**) and slower (**B**) than the median. The mid-green dotted lines show the time points corresponding to mean target onset latency (relative to response), C1 latency, and the emergence of evidence-dependent CPP buildup, demonstrating a delay of approximately 70 ms from the C1 to evidence-dependent buildup. The dark green dotted line shows mean target onset latency in hard blocks. Error shading indicates standard error of the mean taken from N=18 participants.

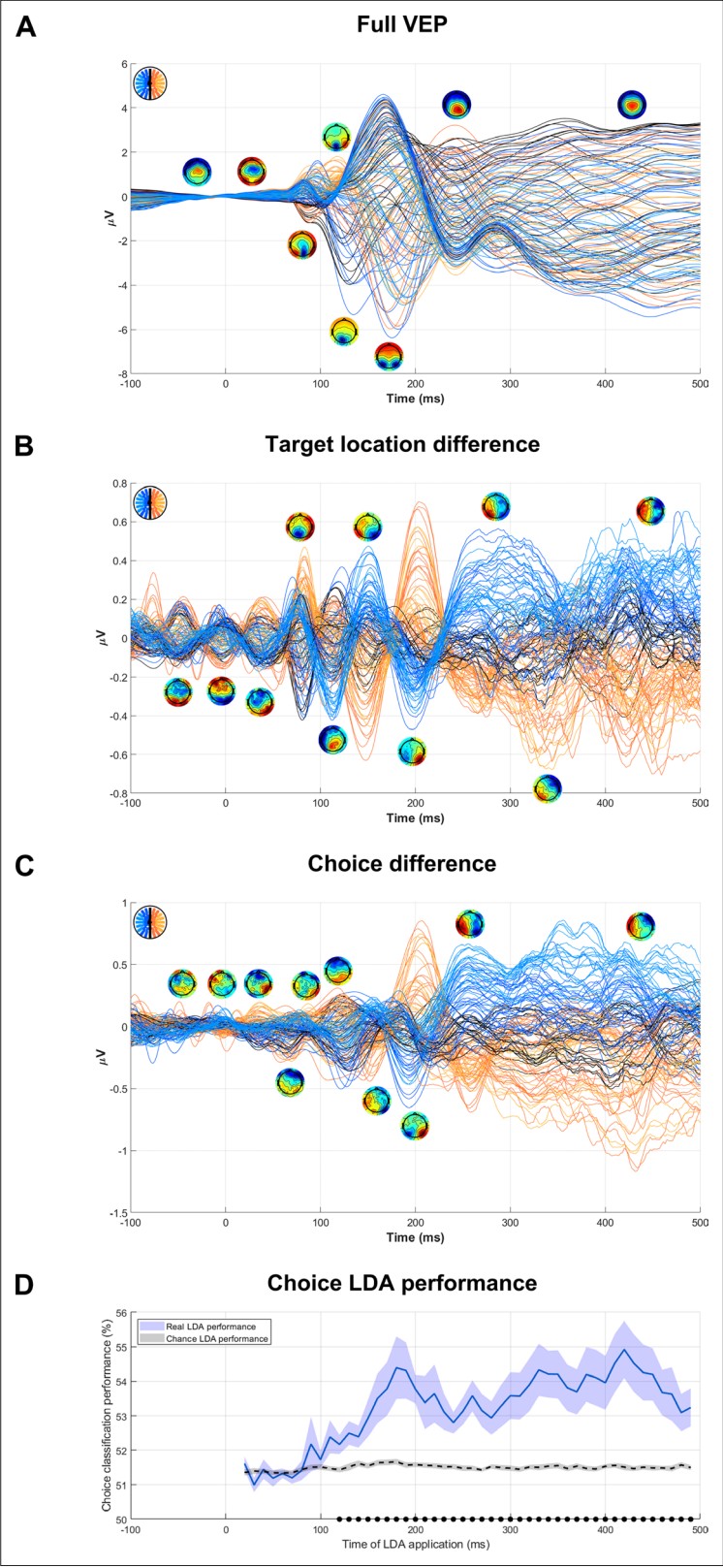

**Figure 5.** Full target-evoked waveform along with components predicting choices and physical evidence. (**A**) Butterfly plot of the target-evoked VEP in hard blocks showing topographies of the main components observed. (**B–C**) As above but for the subtraction of lower-target trials from upper-target trials (**B**) and the subtraction of chose-lower trials from chose-upper trials (**C**). Note that the scale of the y-axis in these panels is smaller than in

*Figure 5 continued on next page*

*Figure 5 continued*

panel A by a factor of 10. (**D**) Percentage of choices in hard blocks accurately classified by linear discriminant analyses (LDA) applied to choice outcomes. Each classifier was trained on a 10 msec window and then applied to that same window, separately for each participant. To account for overfitting, we shuffled choices and refitted the LDA to demonstrate chance performance (dotted black line). The black dots at the bottom indicate time windows where the real classifier performed significantly better than chance. Error shading indicates standard error of the mean taken from N=18 participants.

analysis to map the strength of choice-related activity across electrodes over time. This demonstrated that scalp activity became increasingly choice-predictive, substantially exceeding the level seen in the C1 alone, up to approximately 180 ms and then plateaued (*Figure 5D*).

## Discussion

By demonstrating choice probability in the C1 component of the VEP, this study showed that initial afferent V1 activity can predict perceptual choice variations above and beyond the physical evidence itself. As well as being unexplained by the physical stimulus, the effect was also unaccounted for by choice history, attention, or any other choice-predictive signal that we could find in the EEG. This reduces the likelihood that the effect was mediated by another choice-predictive variable and bolsters the conclusion that it reflected readout of initial afferent V1 activity. The effect was limited to an intermediate RT range of approximately 230 ms to 400 ms, which is also indicative of decision readout because choices that are informed by a transient signal ought to be made in a timeframe consistent with the timing of that signal. In fact, this RT-contingency would be difficult to explain if the effect were mediated by attention or some other baseline cognitive factor. Such an effect should be associated with the fast and slow tails of the RT distribution where pre-existing biases would tend to exert greatest influence (*Hanks et al., 2011*; *Ratcliff et al., 2016*). An effect mediated by visual signals further along the analysis pipeline could, however, account for the RT contingency, and although several such signals were controlled for in our analysis, we cannot account for visual activity that is inaccessible with EEG. Nevertheless, the multitude of factors controlled for, and the fact that this led to an increase, not a decrease, in the magnitude of the C1 choice probability effect, renders readout of initial afferent V1 activity the best account for these results.

This finding contributes to the debate about the degree of separation of early visual processes from cognitive influences (*Crick and Koch, 2003*; *Hillyard and Anllo-Vento, 1998*; *Hochstein and Ahissar, 2002*; *Lamme and Roelfsema, 2000*; *Leopold, 2012*; *Pylyshyn, 1999*; *Tong, 2003*; *Vetter and Newen, 2014*), arguing against accounts of early V1 activity as a purely automatic conduit of information that is 'cognitively impenetrable' (*Pylyshyn, 1999*). Not only was the C1 read out for decisions in this task, but choice history also had a bearing on its amplitude, showing that initial afferent V1 activity both influenced and was influenced by cognitive processes (decision making and bias, respectively). Interestingly, choice history seemed to exert separate influences on the C1 and behaviour as the effects were associated with different RT ranges and operated in opposite directions - the C1 bias was congruent with the previous choice whereas the (dominant) behavioural bias was to switch responses. The simultaneous operation of two countervailing biases has been observed in other contexts, such as in speeded choices in a value-biased random dot motion task (*Corbett et al., 2023*), and in other choice history analyses (e.g. *Urai and Donner, 2022*). Another intriguing aspect of this choice history bias of the C1 was its restricted expression to an intermediate range of RTs similar to the core choice probability effect, which is remarkable because the C1 bias precedes the instantiation of RT in a given trial. There was no evidence that these effects were mutually dependent as each choice coefficient remained unchanged regardless of whether the other regressor was included in the model, and indeed, as outlined above, the C1 and behavioural biases operated in opposite directions. Therefore, to explain the RT contingency for the C1 choice-history bias, we speculate that the speed-accuracy trade-off could fluctuate from trial to trial and that the corresponding decision bound fluctuations (*Heitz and Schall, 2012*) could be implemented by pre-determining decision weights across visual areas. For example, to achieve faster decisions, the sensory evidence requirement could be reduced by placing greater emphasis on initial afferent V1 evidence. In such a case, the RT contingency of the above choice history bias could be explained if the C1 bias is exerted in proportion with

the planned emphasis of C1 evidence for the upcoming decision. Regardless of whether this account is true, the existence of choice history influences on the C1 argues against accounts that initial afferent V1 is shielded from cognitive influences.

The question of cognitive influences on initial afferent V1 activity need not, however, be one of a dichotomy. Indeed, choice prediction was weaker in the C1 than in later signals, mirroring the greater effects in later signals of both spatial attention (*Treue, 2001*) and of choice probability in the animal literature (*Nienborg et al., 2012*). It therefore remains apparent that such early stages of vision are influenced by cognition to a lesser degree than later stages, or under more restricted circumstances. For example, the C1 component has often been regarded as a low-level visual signal that is not subject to top-down cognitive influences. This owes in great part to many failed attempts to find a modulation of the C1 by selective attention, while modulations of later signals are routinely found (*Hillyard and Anllo-Vento, 1998*). Although a recent meta-analysis found that such effects are in fact present despite the many null results (*Qin et al., 2022*), some studies suggest that a critical determining factor for observing such effects is whether the task demands imposed by the configuration of stimuli warrant the modulation (*Dassanayake et al., 2016*; *Fu et al., 2010*; *Mohr et al., 2020*; *Rauss et al., 2012*), suggesting that top-down influences on early vision are not observed routinely but are instead restricted to specific scenarios that require them. This is consistent with findings from the animal literature that V1 modulations by spatial attention are stronger in the presence of stimulus competition (*Motter, 1993*). In a similar vein, *Kang and Maunsell, 2020* demonstrated that choice probability in V1 neurons could be increased by changing the spatial frequency of targets to be more aligned with V1 tuning curves, showing that choice probability strength is sensitive to the task utility of visual areas. In the present study, one might have expected that the stimulus configuration would have made V1 less useful because the task required contrast to be aggregated over large regions of visual space and V1 receptive field sizes are small (*Dumoulin and Wandell, 2008*) and horizontal connections are slow (*Girard et al., 2001*). However, this aggregation could be achieved by convergence of parallel outputs from V1 onto the same decision process without the need for local aggregation in V1. Moreover, a key task demand that likely enhanced the relative utility of initial afferent V1 activity was the tight deadline, which directed the speed-accuracy trade-off (SAT) in favour of speed, thereby favouring earlier visual representations.

Another interesting aspect of the results of this study was that C1 choice probability was associated with a smaller range of RTs than above-chance performance. This was clearest at slow RTs of around 400 ms where the conditional accuracy function peaked, yet the C1 was no longer choice predictive. This is consistent with the fact that sensory evidence would have continued to be elaborated through downstream visual areas whose readout would dilute the influence of the short, transient C1. What was less clear was whether the C1 accounted for the fastest above-chance decisions. Although there was a trend for above-chance performance to emerge at faster RTs than C1 choice probability, the difference approached, but did not reach, statistical significance, making it difficult to draw a firm conclusion. It is therefore worthwhile to consider interpretations of both scenarios. Given that the C1 index initial afferent V1 activity (*Clark et al., 1994*; *Di Russo et al., 2002*; *Jeffreys and Axford, 1972*; *Mohr et al., 2024*), it would seem natural for its choice probability to account for the fastest above-chance responses. What requires more explanation is the possibility that it didn't. This scenario purports that there is an earlier visual representation that could account for the fastest responses. One potential contender is the magnocellular pathway, which is thought to provide a quick, low-resolution picture of the visual scene in order to provide timely top-down modulation of early computations in the more detail-oriented parvocellular pathway (*Bar, 2003*; *Kaplan and Shapley, 1986*; *Kveraga et al., 2007*; *Laycock et al., 2007*). Indeed, previous research has found that C1 response properties better match those typical of the parvocellular pathway than the magnocellular pathway (*Foxe et al., 2008*; *Lalor and Foxe, 2009*; *Murphy et al., 2012*), although there is some difficulty in using this method to distinguish the two pathways (*Skottun, 2015*; *Skottun and Skoyles, 2011*). It is worth noting that the spatial aggregation required in the present task was well suited to the large receptive field sizes of the magnocellular pathway (*Kaplan and Shapley, 1986*), making its involvement in informing the fastest decisions seem plausible.

Another goal of this study was to estimate the overall timeline of events in the decision formation process. The C1 choice probability effect estimates an upper limit for the onset of sensory evidence readout of 80–90 ms. The latency at which this began to impact decision formation was estimated

at approximately 150 ms by the onset of evidence-driven buildup in the CPP, a delay of 60–70 ms. A similar delay of approximately 50 ms between the decision and motor stages was estimated by the RT at which above-chance accuracy first emerged (200ms). Although there was considerable uncertainty in these latency estimates, they converge on a transmission delay in the order of tens of milliseconds between stages of the choice formation process. This delay is long when compared with typical transmission delays between cortical areas, which are usually less than 10 ms (*Innocenti et al., 2014*; *Nowak and Bullier, 1997*; *Tomasi et al., 2012*), possibly indicating the involvement of slower local computations (*Girard et al., 2001*), but notably they are in line with recent delay measurements in the propagation of decision variables across different cortical areas in multi-region recordings in rats (*Bondy et al., 2025*).

To the best of our knowledge, this study is one of only a few to have used non-invasive neurophysiology to measure choice probability in a sensory signal. In doing this, we take the critical step of studying choice probability in humans, which is important to account for possible species differences in decision readout strategy and also to offset the possible influence of the extensive task training that animals necessarily receive, which can modulate sensory signals in humans, especially with extensive task exposure (*Bao et al., 2010*; *Walsh et al., 2024*). Other human studies (*Amitay et al., 2013*; *Waschke et al., 2017*) have used EEG to predict which of a sequence of identical auditory tones would be chosen as the 'odd one out', finding choice-related activity in the P1-N1 peak in the former study and stimulus-induced theta-band activity in the latter. *Vilidaite et al., 2019* found choice-predictive activity from approximately 100 ms in a two-interval contrast comparison task. Although they did not find choice probability in the C1, they did not actively seek it and instead used a machine learning approach, which similarly did not uncover the C1 effect in the present study (*Figure 5D*), presumably due to that analysis not accounting for a potentially restricted RT range for the effect, combined with the absence of prior knowledge of systematic topographic behaviour of the C1 input in the machine-learning approach (see Methods). Moreover, none of these studies controlled for other choice-predictive signals as was done here. *Wilming et al., 2020* used MEG to disentangle feedforward from feedback processing in a contrast discrimination task, not unlike what was done here. Unlike here, they found no stimulus-independent choice probability in V1 activity (as measured by visual evoked gamma activity). However, since their primary goal was to disentangle feedforward from feedback visual processing, they employed a protracted decision paradigm with a delayed report. Therefore, if one is to draw a comparison between their results and ours, the comparison ought to be made for slow response time trials, where C1 choice probability was also not present here. Thus, these experiments are not, in fact, contradictory but rather serve to highlight the relative impacts of feedforward and feedback processing on perceptual choices in different contexts. The present experiment suggests that feedforward activity contributes to the choice when it is made quickly based on discrete evidence, while the data of *Wilming et al., 2020* suggest that feedback processing may be more important in scenarios where sensory evidence is accumulated over a longer time. An interesting question for future research will be to explore the determinants of this balance between feedforward and feedback-driven choice formation.

In this study, we demonstrated that initial afferent V1 activity can predict speeded contrast judgements and argued that this likely reflected decision readout, ascribing a degree of cognitive flexibility to one of the earliest stages of visual processing. However, this study has limitations that are important to highlight. Firstly, although we controlled for as many choice-predictive signals as we could, there is no guarantee that all relevant signals can be isolated in EEG. Responses across the myriad individual areas of the visual cortex sum together on the scalp and become difficult to separate in EEG. Indeed, V1 is perhaps the only individual visual area with a claim to dominate a VEP component, the C1, with the subsequent P1 component likely reflecting a large collection of extrastriate areas (*Di Russo et al., 2002*). Therefore, it remains possible that the C1 choice probability effect that we found was mediated by sensory activity that we could not measure. Secondly, although the primary reason to include two response deadlines in this study was to generate a wide RT distribution, another reason was to test for sensitivity in afferent V1 readout as a function of speed pressure. We found no such difference here, but the RT range associated with C1 choice probability lay comfortably within the RT distributions of both regimes, suggesting that the choice of deadlines was not well suited to test this particular question. A follow-up study might therefore include a more liberal deadline to address the sensitivity of V1 weighting to speed pressure in contrast judgements. Finally, the scope of this

study was limited to speeded contrast judgements, while the bulk of decision-making tasks involve more protracted evidence and deliberation periods. Therefore, it remains possible that early visual representations do not usually contribute, or contribute to a much smaller relative extent, to decision readout under more relaxed speed requirements, which will be an interesting question for future work. Our study provides a useful template for such studies, demonstrating how choice probability measurements can be combined with RT-based arguments and statistical control of other choice-predictive signals to infer direct readout of sensory responses for decision formation. In so doing, we hope to inform future work seeking to probe the evidence accumulation strategies used in decision making across varying task settings.

## Methods

### Participants

Twenty healthy young adults aged 25.6±3.9 (M ± SD; 10 Females, 17 right-handed) took part in the experiment. After providing informed written consent on the first day, participants attended three sessions. The first was behavioural training, with EEG recordings taking place on the latter two. If no clear C1 component was identifiable at the second session, the experiment was terminated and the participant did not attend the third session, which was the case for two participants, leaving a final sample size of N=18. Participants were paid €20 for the first session and €30 for each of the other two sessions. All participants had normal or corrected-to-normal vision and reported no neurological or psychiatric conditions. All operations were approved by the Human Research Ethics for Sciences board of University College Dublin and adhered to the guidelines set out in the Declaration of Helsinki.

### Task

Participants carried out a difficult contrast discrimination task. Two arrays of gratings appeared simultaneously for 39 ms in the upper left and lower right quadrants of the visual field (see *Figure 1*) and participants determined which array had greater net contrast, indicating their response with a left/right hand response on the up/down arrow key of a keyboard. This mapping was counterbalanced across blocks. To instil speed pressure and elicit a wide range of response times, two deadline regimes were implemented (400 ms and 600 ms), and participants met these on most trials (see *Figure 3B*). The deadlines were implemented on separate days to minimise any carry-over of strategy, with the order counterbalanced between participants. A brief 800 Hz tone sounded at 800 ms post-stimulus to inform participants whether they were on time (180 ms duration) or late (380 ms duration). With an overall trial duration of 1,960 ms, this left a minimum of 780 ms following feedback offset until the next trial. This inter-trial interval (ITI) was fixed to avoid introducing inter-trial variability that may risk producing a confound. Although this carries the risk of facilitating slow adaptation effects (*Baccus and Meister, 2002*), such an effect would be more constant across trials and therefore would not be a confound. Participants were not provided with accuracy feedback on individual trials. Instead, they saw this summarily at the end of each block as a percent correct figure, along with how often they responded on time, how often they chose the upper-field array, and a performance score for that block, calculated by awarding 10 points for every correct, timely response and subtracting 7 points for all other trials. To incentivise participants to respond within the deadline, they were instructed to strive for as high a score as possible.

### Stimuli

Stimuli were presented in a dark, sound-attenuated chamber on a 1024×768 Dell E771p CRT monitor (32.5×24.5 cm, 75 Hz refresh rate) at a distance of 57 cm from the participant who was seated at a chin rest. Participants maintained a stable gaze on a white fixation square (0.23° × 0.23°), which was displayed against a uniform grey field of 64 cd/m² throughout the experiment. Stimuli consisted of two arrays of achromatic circular gratings that flashed simultaneously in the upper left and lower right visual quadrants for 39 ms (see *Figure 1A*). Each grating had a spatial frequency of 5 cpd, was oriented vertically, and had a radius of 0.33°, with a centre-spacing of 0.79° both radially and along the azimuth. Both arrays contained a 9×9 grid (eccentricity by polar angle) of such gratings, ranging from 6° to 13° of visual angle radially and subtending an annular segment of 70° polar angle (ranging from 100° to 170° for the upper left array and 280° to 350° for the lower

right array). We designed the stimuli in this way for several reasons. First, since the C1 has opposite polarity for the upper and lower visual field and its amplitude scales with stimulus contrast (*Gebodh et al., 2017*), using an upper and lower field quadrant produced a composite signal of competing positive and negative polarity that could be used as an index of differential contrast (see *Figure 1B*). Second, since analyses would focus on this difference signal, which we expected to be small and which relied on a strong overlapping C1 topography being present for both the upper and lower visual field, robust signal-to-noise ratio was a principal concern. Spanning stimuli across large segments of the visual field produces very large C1 components (*Pourtois et al., 2008*; *Rauss et al., 2009*). Moreover, the arrays are positioned such that most gratings should produce approximately midline topographies as they avoid regions surrounding the horizontal and vertical meridians where C1 topography changes the most (*Mohr et al., 2024*). Together with the large coverage, this helped to ensure relatively consistent topographies across individuals, which are usually quite variable (*Mohr et al., 2020*; *Vanegas et al., 2013*). Third, since alpha-band lateralisation can be used to assess relative attentional engagement between the left and right visual fields (*van Dijk et al., 2008*; *Foxe and Snyder, 2011*; *Kelly et al., 2006*; *Klimesch, 2012*; *Thut et al., 2006*), using a left and right field quadrant allowed for measurement of any pre-stimulus attentional biasing between the two arrays.

The participant's task was to indicate which of the two arrays had greater mean contrast, defined as the 'target'. Across all gratings, average contrast was 60%, but with the average of one array incremented and the average of the other decremented by the same amount (DC), which was titrated to each participant to achieve approximately 55% accuracy (see Difficulty Titration section below). To ensure that participants made use of all gratings to inform their decisions, contrast also varied across gratings within each array. The residuals of these deviations were constrained to sum to zero so that the overall DC did not vary across trials. To achieve this, we took 9 equally spaced values in the range of ±20% (5% intervals), produced 9 sets of these 9 values to form a 9x9 grid, shuffling within each column, and taking a random permutation of these sets in each trial. This gave the appearance of a random spatial distribution while keeping the same marginal distribution of deviations across polar angle and eccentricity for each trial. Although participants could, in principle, perform the task accurately by focusing on a single arm of polar angle (since each one contained the full information of the array), it is unlikely that participants were able to notice this regularity as the stimuli appeared so briefly.

## Procedure

In the first testing session, difficulty was titrated to yield relatively stable performance at approximately 55% accuracy (see Difficulty Titration section below for details). This difficulty level was then kept constant for the main task in the following two testing sessions while EEG and eye tracking were recorded. In each EEG session, 10 blocks of 120 trials were completed with a given deadline, which yielded 600 trials for each combination of target location and deadline regime. Due to the titration to 55% accuracy, there were almost as many error trials as correct trials, thus enabling powered analyses related to choice. The counterbalancing of the mapping between left/right hand response and upper/lower target meant that lateralized motor preparation signals tended to cancel overall, minimising interference with C1 measurements. To minimise cognitive burden, this mapping was not changed frequently, swapping halfway through each session and between the two days, with the mapping on the first day counterbalanced across participants.

The differential C1 response of interest, produced by competing positive and negative polarity responses from the lower and upper arrays, was low in amplitude. Therefore, to assist signal identification for the main data, two easy blocks of 160 trials were administered prior to starting the main task to yield a strong signal that could be used to identify C1 topography for each participant. In the first such block, an easy version of the main task was carried out, where DC was set to 30%, which yielded a strong differential C1 signal and ceiling performance for all participants. In the second block, only the target array appeared, and the participant simply indicated its location. Upon completing these blocks, the experimenter inspected the EEG data to identify the C1 components for the upper- and lower-field separately, and if either of them was absent, the experiment was ended at that stage.

## Difficulty titration

Task difficulty was individually titrated to approximately 55% accuracy in order to yield roughly equal trial numbers across all combinations of choice outcome and target location while still keeping participants engaged in the task. To account for any response bias (either perceptual or motor), we also titrated a bias parameter to counteract the bias, which was a term added to DC to make the task harder/easier depending on the direction of the target. Although titrating these parameters is not actually necessary to obtain a valid choice probability measurement, which is simply the correlation between fluctuations in choice and fluctuations in neural activity for a given target, doing so helped to facilitate well-powered choice analyses by ensuring balanced trial counts across all possible response bins.

To carry out this titration, the experimenter iteratively adjusted the two parameters as participants completed short blocks of the task, inspecting choices by target location after each block. If overall accuracy was too high, DC was decreased and vice versa. If too many of one choice were made, the bias parameter was adjusted to counteract this. Participants also improved at the task as the titration blocks went on, requiring the experimenter to make further parameter adjustments. Many participants hit a saddle point in their learning during titration where performance was stable for a number of blocks but then began to improve again. Therefore, once the first saddle point was reached, the experimenter repeated the same parameters for a number of blocks to check if the participant would start a new wave of improvement. If this happened, titration was stopped after performance stabilised a second time; otherwise, it was stopped once the experimenter had the impression that a second wave of improvement seemed unlikely. At this point, the DC and bias parameters were set and remained unchanged for the main experimental blocks in the subsequent EEG sessions.

## Data acquisition and data processing

EEG data was recorded at 512 Hz by an ActiveTwo Biosemi system with 128 scalp electrodes following the Biosemi ABC layout (Biosemi, The Netherlands) and six external flat-faced electrodes placed above and below the left eye, on the left and right outer canthi, and on the left and right mastoids. Eye gaze and blinks were monitored via the four electrooculograms (EOG) mentioned above and using an Eyelink Plus 1000 Tower system (SR Research, ON, Canada) recording at 1000 Hz. EEG processing was carried out using a combination of in-house Matlab scripts (Mathworks, The United Kingdom) and EEGLAB routines (*Delorme and Makeig, 2004*). Continuous data were low-pass filtered by convolution with a 77-tap Hanning-windowed sinc function that provided a 3 dB corner frequency of 35.3 Hz and 83.5 dB of attenuation at 50 Hz (the mains frequency). Noisy channels were identified by visual inspection and interpolated using EEGLAB's *eeg_interp* function, after which the data were re-referenced to the average of all scalp channels.

At this point, target-locked epochs were extracted from –600 ms to 400 ms and baseline corrected to a window of –50 ms to 30 ms. These were then screened for artifacts using thresholds chosen by visual inspection. Blinks were identified both from the eye tracking data and from the difference between the vertical EOG channels by applying a threshold of 40 µV (the 50ms before and after this violation were also considered part of the blink). Slow wave drift and muscle artifact were both identified by thresholds of power ratios between different frequency bands. Slow wave drift was defined as a ratio exceeding 5:1 between 0–3 Hz and 3–7 Hz and muscle artifact was defined as a ratio exceeding 2:1 between 20–40 Hz and 3–7 Hz. To identify any further artifacts, we applied a threshold of 50 µV to voltage throughout the epoch, and a threshold of 40 µV to its range during the baseline window. If any of these artifacts were observed, the offending channels in that epoch were excluded from analysis. In the case of blinks, the time segment when the blink happened was excluded, but the rest of the epoch was retained (i.e. whether or not the trial was excluded from an analysis depended on whether the signal used in the analysis coincided in time with the blink). However, when blinks coincided with stimulus presentation, the trial was rejected in full. Trials were also rejected in full if, within 100 ms surrounding stimulus onset, a saccade took place or median gaze deviation exceeded 1° of visual angle. Average trial-loss rates were 9.8% (SD = 4.3%).

## Data analysis

The data reported in this manuscript as well as the analyses performed have been made publicly available on an online repository: here.

## C1 measurement

Because the stimulus was designed to generate two opposing C1s from each of the two possible target arrays, C1 measurement electrodes were determined based on the block of easy trials where the large contrast difference between arrays elicited an easily identifiable C1 even with signal cancellation. This was done on a single subject basis to account for individual differences in C1 topography. Trials were averaged between 80 and 90 ms for each target location separately and subtracted. This was plotted as a topography to find the electrodes that showed amplitude differences between upper and lower target locations. This difference in topography had a positive and negative pole (see *Figure 1B* for the grand average) because upper-field and lower-field C1s tend to have slightly different topographies (*Mohr et al., 2024*). Therefore, to maximise the robustness of our C1 measurement, we measured it as the difference between the strongest electrodes from both poles such that deflections in the positive direction were consistent with lower-field targets. Having chosen measurement electrodes, they were subsequently applied to measure the C1 as a single scalar variable on a single-trial basis in hard blocks.

## Mediating signals

To control for signals that might mediate choice probability in the C1, occurring either before or after, we performed a comprehensive search for any choice-predictive activity in the EEG from 400 ms before targets to 600 ms afterwards, both in the time domain and time-frequency domain. To convert to the time-frequency domain, we used a Short-time Fourier Transform (STFT) with a window length of 400 ms and step interval of 25 ms, taking the magnitude of the complex Fourier values and dividing by half the number of time samples. With data represented in both domains, participant averages were taken for both, as a function of choice, and subtracted, a difference that deviates from zero when neural activity is predictive of choices. This was then averaged across participants and plotted for visual inspection to identify time windows that stood out as choice predictive, retaining the ones that passed a t-test. Although this approach can lead to false positives (*Luck and Gaspelin, 2017*), it was more important here to be comprehensive and avoid false negatives than to avoid false positives, given the goal of this analysis to identify variables to control for in subsequent choice analyses. Although we also present these choice predictive signals in the results, we do so simply to provide a reference for future studies involving tasks with contrast comparisons across space.

The relevant time-domain plot is shown in *Figure 2A*. For the time-frequency domain, similar plots were first generated for every non-zero frequency up to 30 Hz and choice signals at adjacent frequencies were then combined if they had mostly overlapping time windows and electrodes, being subjected to t-tests after that point. The final time-frequency choice signals are shown in *Figure 2B*. By doing this, we identified seven signals in the time domain and six in the time-frequency domain (see Results).

## Statistical analysis

The majority of analyses were carried out using mixed-effects models with random intercepts, treating single trials as observations and participants as the grouping variable. One exception to this was an analysis of response time variability since this is not defined at the single trial level, so paired samples t-tests were used in this case instead. In the mixed-effects models, the regression was either a logistic one, if predicting a binary variable such as choice outcome, or a linear one, if predicting a continuous variable such as the C1 or RT. Continuous variables were converted to z-scores except for RT which was left in units of seconds. For binary variables, a code of 1 was assigned to upper-field targets and choices, correct responses and the 600 ms deadline condition, while a code of 0 was assigned to lower-field targets and choices, incorrect responses and the 400 ms deadline. Trials were omitted from analysis if they either did not pass the rejection screening described previously or if RT was above 600 ms, the latter of which was only true in 0.7% of trials.

Our main hypothesis was that C1 amplitudes would be predictive of choices within a specific range of response times (RTs) and that this relationship would not be mediated by any other choice-predictive variable or indeed by physical stimulus differences between target locations that were also choice predictive. These two points, the absence of mediation and RT dependence, were addressed by including as many choice-predictive covariates to control for as possible and including an interaction with a quadratic function of RT for all of them. This meant that three

regressors were included for each choice predictor, one for the main effect and one each for the interactions with RT and its square. The choice-predictive covariates included the signals identified in the comprehensive search described above, as well as the true target location, which we know correlates with both choices and the C1, and the previous trial's choice, which has been shown to influence both choices and sensory signals in upcoming trials (*St John-Saaltink et al., 2016*; *Macke and Nienborg, 2019*). By including all these covariates alongside one another in a single model, all with the same RT interactions to allow for RT-dependent choice relationships, a C1 choice relationship that was mediated by these variables would be accounted for. However, a C1 choice relationship that was at least partially independent of these covariates would remain despite their inclusion.

In the event of a significant interaction between the C1 and the quadratic function of RT, the nature of the RT-dependent relationship was visualised by applying a sliding RT window procedure where the same model as above was evaluated in each window but without the RT interaction terms so that the C1 coefficients could be plotted as a function of RT window, multiplying by −1 in plots so that positive values would be associated with choice-congruent C1 amplitude. We applied windows based on RT percentiles rather than using simple RT windows of constant width so that analyses at fast and slow RTs would not be less powered than those at intermediate RT. The width of the window was 20% and was centred at 10% up to 90% in 1% increments. Note that random intercepts were excluded from these models because otherwise the model fits would not converge at the fastest and slowest RT windows where trial counts across participants were variable. We re-evaluated the original model without random intercepts to confirm that this was not critical to the model's performance and the results were not qualitatively different. To estimate uncertainty in the significant RT range for C1 choice probability, a bootstrapping procedure was used with 10,000 samples in which participants were re-sampled with replacement from the original participant pool. In each sample, we determined its RT range and estimated the beginning and end of it by finding the windows where the C1 parameter crossed zero. To determine the RT range initially, we modelled the sequence of significant/non-significant test outcomes across RT windows as a single unbroken sequence of significant windows with non-significant windows before and after, finding the sequence with the highest number of windows in agreement with the true sequence. This yielded a distribution of onset RTs and offset RTs for the significant RT range, which we used to generate 95% confidence intervals. The same approach was used to estimate the RT at which above-chance performance emerged, except that the model evaluated at each RT window was a simple intercept model to predict correct/erroneous responses. To compare this to the RT onset of C1 choice probability, both models were evaluated in each bootstrap sample.

## Measuring the onset of evidence-dependent buildup in the CPP

Although the primary aim of the study was to assess choice probability in the C1, we also measured the onset of evidence-dependent buildup in the centroparietal positivity (CPP) to place this analysis within the context of the broader timeline of choice formation. The CPP is a signal that has been tightly linked with decision making - it starts to build up following the onset of decision-related sensory evidence, its buildup rate scales with the strength of sensory evidence, it peaks prior to response, and it is observed regardless of the need for an immediate motor response (*Kelly and O'Connell, 2013*; *O'Connell et al., 2018*; *Twomey et al., 2016*). To choose a measurement electrode for this signal, we averaged across all trials and plotted the topography from −150 ms to −50 ms before responses, choosing the electrode in the centroparietal region with the highest amplitude. This was electrode 4 in Biosemi's ABC layout, lying between standard 10–20 sites CPz and Pz.

To measure the onset of evidence-dependent buildup in the CPP, we measured its slope over time and compared this between hard and easy blocks to find the time point at which the slope first became steeper in the easy block. To do this, we applied a sliding 50 ms time window in intervals of 10 ms and found the line of best fit, conducting a paired t-test between easy and hard blocks for each window. The onset of evidence-dependent buildup was estimated as the first window where the slope was significantly higher in the easy block and remained so up to its peak. To capture uncertainty in this estimate, we generated 10,000 bootstrap samples of participants by re-sampling them with replacement and carrying out this procedure in each sample to generate a distribution of onset latency estimates. We carried out this procedure on the response-locked waveform to avoid overlapping components

associated with target onset that would obscure the onset of the CPP. Therefore, to convert our onset latency estimate to a delay from target onset, we subtracted it from mean RT.

## Additional information

### Funding

| Funder | Grant reference number | Author |
|---|---|---|
| Irish Research Council | GOIPG/2016/123 | Kieran S Mohr |
| Wellcome Trust | 219572/Z/19/Z | Simon P Kelly |
| Irish Research Council | GOIPD/2024/870 | Kieran S Mohr |

The funders had no role in study design, data collection and interpretation, or the decision to submit the work for publication. For the purpose of Open Access, the authors have applied a CC BY public copyright license to any Author Accepted Manuscript version arising from this submission.

### Author contributions

Kieran S Mohr, Conceptualization, Data curation, Software, Formal analysis, Funding acquisition, Investigation, Visualization, Methodology, Writing – original draft, Project administration, Writing – review and editing; Simon P Kelly, Conceptualization, Resources, Supervision, Funding acquisition, Investigation, Methodology, Writing – review and editing

### Author ORCIDs

Kieran S Mohr ⓘ https://orcid.org/0000-0002-6165-7810
Simon P Kelly ⓘ https://orcid.org/0000-0001-9983-3595

### Ethics

Participants in this study provided written informed consent to take part in the study and for the data to be published. All procedures carried out were approved by the Human Research Ethics Committee for Sciences at University College Dublin.

Reviewer #1 (Public review): https://doi.org/10.7554/eLife.109046.3.sa1
Reviewer #2 (Public review): https://doi.org/10.7554/eLife.109046.3.sa2
Author response https://doi.org/10.7554/eLife.109046.3.sa3

## Additional files

### Supplementary files

MDAR checklist

### Data availability

The data used in this study can be downloaded from the following OSF link: https://osf.io/5rtqz/files/osfstorage. This is part of the following OSF project: https://doi.org/10.17605/OSF.IO/5RTQZ.

The following dataset was generated:

| Author(s) | Year | Dataset title | Dataset URL | Database and Identifier |
|---|---|---|---|---|
| Mohr K, Kelly S | 2025 | Choice Probability in the C1 component of the Visual Evoked Potential | https://osf.io/5rtqz | Open Science Framework, 5rtqz |

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
