## [Editor Report · eLife Assessment]

This **important** study reports that EEG recordings of the earliest stage of information processing in human visual cortex can be used to predict subsequent choice responses. The findings provide novel, **convincing** evidence for integrative processing in low-level sensory cortices at the level of scalp-recorded potentials, with the exact nature of the neural signals at the single cell level to be determined. The paper is likely to be of interest to neuroscientists interested in the contribution of early sensory signals to decision making.

---

## [Referee Report · Reviewer #1 (Public review)]

General assessment of the work

In this manuscript, Mohr and Kelly show that the C1 component of the human VEP is correlated with binary choices in a contrast discrimination task, even when the stimulus is kept constant and confounding variables are considered in the analysis. They interpret this as evidence for the role V1 plays during perceptual decision formation. Choice-related signals in single sensory cells are enlightening because they speak to the spatial (and temporal) scale of the brain computations underlying perceptual decision making. However, similar signals in aggregate measures of neural activity offer a less direct window and thus less insight into these computations. The authors do a good job justifying their focus on the C1 component and illustrating how it may behave under different simulated scenarios. The results are interesting, although it is difficult to specify which reasonable hypothesis is exactly ruled out by these results. One interpretation is that V1 activity directly guides perceptual decisions in this task. Alternatively, higher-level areas may do this, provided that their activity largely reflects their V1-inputs. This certainly seems possible in a simple task like this.

Summary of substantive concerns

I have no substantive concerns about the revised version of the paper.

---

## [Referee Report · Reviewer #2 (Public review)]

Summary:

Mohr and Kelly report a high-density EEG study in healthy human volunteers in which they test whether correlations between neural activity in primary visual cortex and choice behavior can be measured non-invasively. Participants performed a contrast discrimination task on large arrays of Gabor gratings presented in the upper left and lower right quadrants of the visual field. The results indicate that single-trial amplitudes of C1, the earliest cortical component of the visual evoked potential in humans, predict forced-choice behavior over and beyond other behavioral and electrophysiological choice-related signals. These results constitute an important advance for our understanding of the nature and flexibility of early visual processing.

Strengths:

The findings suggest a previously unsuspected role for aggregate early visual cortex activity in shaping behavioral choices.

The authors extend well-established methods for assessing covariation between neural signals and behavioral output to non-invasive EEG recordings.

The effects of initial afferent information in primary visual cortex on choice behavior is carefully assessed by accounting for a wide range of potential behavioral and electrophysiological confounds.

Caveats and limitations are transparently addressed and discussed.

Weaknesses:

Due to the inherent limitations of scalp-recorded visual evoked potentials, the results cannot be directly compared to invasive recordings in animal models.

---

## [Author Response]

The following is the authors’ response to the original reviews

**Public Reviews:**

**Reviewer #1 (Public Review):**
General assessment of the work:In this manuscript, Mohr and Kelly show that the C1 component of the human VEP is correlated with binary choices in a contrast discrimination task, even when the stimulus is kept constant and confounding variables are considered in the analysis. They interpret this as evidence for the role V1 plays during perceptual decision formation. Choice-related signals in single sensory cells are enlightening because they speak to the spatial (and temporal) scale of the brain computations underlying perceptual decision-making. However, similar signals in aggregate measures of neural activity offer a less direct window and thus less insight into these computations. For example, although I am not a VEP specialist, it seems doubtful that the measurements are exclusively picking up (an unbiased selection of) V1 spikes. Moreover, although this is not widely known, there is in fact a long history to this line of work. In 1972, Campbell and Kulikowski ("The Visual Evoked Potential as a function of contrast of a grating pattern" - Journal of Physiology) already showed a similar effect in a contrast detection task (this finding inspired the original Choice Probability analyses in the monkey physiology studies conducted in the early 1990's). Finally, it is not clear to me that there is an interesting alternative hypothesis that is somehow ruled out by these results. Should we really consider that simple visual signals such as spatial contrast are *not* mediated by V1? This seems to fly in the face of well-established anatomy and function of visual circuits. Or should we be open to the idea that VEP measurements are almost completely divorced from task-relevant neural signals? Why would this be an interesting technique then? In sum, while this work reports results in line with several single-cell and VEP studies and perhaps is technically superior in its domain, I find it hard to see how these findings would meaningfully impact our thinking about the neural and computational basis of spatial contrast discrimination.

We agree that single cell measurements allow for a spatially more detailed analysis, but they are not feasible in humans. Assuming we value insights into the relationship between neural activity and decision making in the human as well as non-human brain, we are restricted to non-invasive measurements such as EEG, which inevitably showcase the neural underpinnings of decision making at a coarser level of analysis. This was the challenge we met with our paradigm design. For example, we chose contrast as the task-relevant stimulus feature in this study because monotonic contrast response functions exist for sensory neurons throughout the visual system, and the aggregated measures that we could attain with EEG would reflect that contrast-sensitivity and hence provide a window onto the encoding of the main decision-relevant quantity. We were specifically interested in initial afferent, contrast-dependent V1 activity reflected in the C1 component (80-90 ms). As we point out in the Introduction, the C1 is unusual among EEG signals in the extent to which it is dominated by a single visual area, V1 (Jeffreys & Axford, 1972; Clark et al., 1994; Di Russo et al., 2002; Ales et al., 2010; Mohr et al., 2024), and even if other downstream areas also make a minor contribution in the C1 time period, it still represents a very low-level sensory response early in the sensory analysis pipeline, appropriate for addressing our primary question of whether such a low-level signal is used in the formation of perceptual decisions. The alternative hypothesis, that early responses are passed over in decision readout, relates to a fundamental debate about whether early sensory responses are separated from cognition. The possibility that late, but not early, representations are correlated with choices does not imply that the later sensory representations are divorced from the earlier ones, only that there is a noise component that is not shared between the two, such as that produced by the ensuing computations that generate the later representations. Instead, a lack of choice probability in early representations would imply that decision readout is selective in where it sources sensory evidence from, with some possible reasons being to maintain high quality standards for sensory evidence or to impose a layer of separation between cognition and sensation.

As the reviewer points out, the animal literature is highly mixed on the topic of choice probability in V1. Even for orientation discrimination tasks where V1 is ostensibly highly suited given the existence of orientation columns in V1, and even when measurements are taken from V1 neurons with good neurometric performance and/or aggregated across a V1 population (Jasper et al 2019), some studies have reported little to no V1 choice probability. If our alternative hypothesis of no EEG-indexed V1 choice probability flies in the face of well-established anatomy and function of visual circuits, then so also do these empirical findings in the animal neurophysiology literature.

Although there are important aspects of choice probability that are accessible in single cell studies but not in EEG (e.g. noise correlations, details of circuit physiology), our EEG measurements tap into the same phenomenon, just at a different level of analysis, i.e. the neural population level. At this level, we have been able to address whether the full body of sensory responses at a particular stage of visual analysis is systematically related to perceptual decision outcomes. Very similar questions are in fact sometimes addressed in the animal neurophysiology literature; for example, Kang and Maunsell (2020) aggregated single-cell choice probability measurements within visual areas to investigate whether choice probability strength at the level of an entire visual area was sensitive to task demands. The global vantage point of EEG comes with the additional benefit of picking up signatures of other potentially mediating processes such as attention and being able to control for them in our analysis. Our human study thus provides a valuable complementary viewpoint alongside animal neurophysiology work in this area.

Summary of substantive concerns:(1) The study of choice probability in V1 cells is more extensive than portrayed in the paper's introduction. In recent years, choice-related activity in V1 has also been studied by Nienborg & Cumming (2014), Goris et al (2017), Jasper et al (2019), Lange et al (2023), and Boundy-Singer et al (2025). These studies paint a complex picture (a mixture of positive, absent, and negative results), but should be mentioned in the paper's introduction.

We thank the reviewer for highlighting these papers bearing on choice-related activity in V1, only two of which we had cited. The three additional studies do indeed lend further support to our description of the complex picture around V1-CP effects in the literature and we have now included them.

(2) The very first study to conduct an analysis of stimulus-conditioned neural activity during a perceptual decision-making task was, in fact, a VEP study: Campbell and Kulikowski (1972). This study never gained the fame it perhaps deserves. But it would be appropriate to weave it into the introduction and motivation of this paper.

We are aware of this paper, and indeed we ourselves have shown steady-state VEP (SSVEP) correlations with timing and selection of decision reports (O'Connell et al 2012; Grogan et al 2023), but SSVEPs do not provide an index of initial afferent V1 activity in the way that the C1 of the transient VEP does. SSVEPs are evoked by a rapid sequence of stimulus onsets, so that activity cannot be attributed to a particular stimulus onset nor its bottom-up latency resolved, and, being a response to an ongoing stimulus, it combines top-down and bottom-up influences from striate and extra striate areas (Di Russo et al 2007). Indeed, in Campbell and Kulikowski (1972) the SSVEP was almost entirely eliminated when the stimulus was undetected. This is in keeping with robust modulations of the SSVEP by spatial attention (Muller and Hillyard 2000). Cognitive influences of this magnitude are never observed in the C1, and in fact are often not observed at all even when later VEP components show robust modulations (Luck et al 2000), which motivated a recent meta-analysis to address the issue (Qin et al 2022). This highlights the important distinction between the earliest transient VEP activity reflecting mainly the initial afferent response in V1, and steady-state sensory activity reflecting a mix of bottom-up and top-down influences across visual cortex. Because of the importance of this distinction, we have added a reference to the above SSVEP papers to the 3rd paragraph of the introduction along with a statement about the distinction.

(3) What are interesting alternative hypotheses to be considered here? I don't understand the (somewhat implicit) suggestion here that contrast representations late in the system can somehow be divorced from early representations. If they were, they would not be correlated with stimulus contrast.

This same conundrum applies to single-cell studies of choice probability. Do studies showing choice probability in V4 but not V1 for example demonstrate that V4 is divorced from V1? In such studies, measurements are typically taken from large representative samples of neurons from both areas with good neurometric performance in both cases and the task often (though not always) involves a target stimulus feature that is encoded in V1 such as orientation. Why then should V4 but not V1 show choice probability when we know the vast majority of input to the visual cortex passes through V1? It must be that feature representation and choice formation are different things with one not inferring the other. This is true for an EEG study as much as it is for a single-cell study.

The alternative hypothesis in our study is that the early sensory responses indexed by the C1 are not directly used in the formation of the perceptual decision at hand. As outlined in our comments above, this does not imply that those early responses are divorced from later responses. Of course, both are correlated with stimulus contrast and so would correlate with each other across changing contrast but this does not necessitate that their noise is correlated when contrast is held constant because new instantiations of noise can be generated by the computations performed at each stage of visual processing. Thus, the interesting alternative hypothesis is that information contained in the sensory representation generated during initial afferent V1 activity is not used directly to form decisions, and instead, decisions are read out from the outputs of computations performed further downstream. Such an outcome, if it had arisen in our data, would have been consistent with a separation between cognition and early visual processing. Instead, our results suggest a certain level of cognitive interfacing at the lowest and earliest cortical levels of visual processing. We have now added text to the Introduction to highlight the distinction between sensory representation and decision readout in order to make the alternative hypothesis clearer.

(4) I find the arguments about the timing of the VEP signals somewhat complex and not very compelling, to be honest. It might help if you added a simulation of a process model that illustrated the temporal flow of the neural computations involved in the task. When are sensory signals manifested in V1 activity informing the decision-making process, in your view? And how is your measure of neural activity related to this latent variable? Can you show in a simulation that the combination of this process and linking hypothesis gives rise to inverted U-shaped relationships, as is the case for your data?

We thank the reviewer for this suggestion of a simulation, which we carried out using the Matlab code. We have also included new Figure 1-Figure Supplement 1 in the revised manuscript.

In our view, sensory signals in V1 are informing the decision-making process in this task from at least as early as the initial afferent response. The main point about C1 latency in relation to the response-time contingency of the choice probability effect is that the more time that elapses without a decision made (and therefore the more additional sensory processing that contributes to the decision), the more diluted is the contribution of the C1 to the decision by contributions from later representations, and thus choice probability reduces. Likewise, when response times are too quick for C1 evidence to contribute, choice probability is also absent, hence the inverted-U-shaped curve. Moreover, if the C1-choice correlation is mediated by a top-down factor such as attention rather than readout, the inverted-U-shaped curve is not expected because in such a case the relative timing of the C1 and choice commitment would not be relevant.

**Reviewer #2 (Public review):**
Summary:Mohr and Kelly report a high-density EEG study in healthy human volunteers in which they test whether correlations between neural activity in the primary visual cortex and choice behavior can be measured non-invasively. Participants performed a contrast discrimination task on large arrays of Gabor gratings presented in the upper left and lower right quadrants of the visual field. The results indicate that single-trial amplitudes of C1, the earliest cortical component of the visual evoked potential in humans, predict forced-choice behavior over and beyond other behavioral and electrophysiological choice-related signals. These results constitute an important advance for our understanding of the nature and flexibility of early visual processing.Strengths:(1) The findings suggest a previously unsuspected role for aggregate early visual cortex activity in shaping behavioral choices.(2) The authors extend well-established methods for assessing covariation between neural signals and behavioral output to non-invasive EEG recordings.(3) The effects of initial afferent information in the primary visual cortex on choice behavior are carefully assessed by accounting for a wide range of potential behavioral and electrophysiological confounds.(4) Caveats and limitations are transparently addressed and discussed.

We would like to thank the reviewer for these positive remarks.

Weaknesses:(1) It is not clear whether integration of contrast information across relatively large arrays is a good test case for decision-related information in C1. The authors raise this issue in the Discussion, and I agree that it is all the more striking that they do find C1 choice probability. Nevertheless, I think the choice of task and stimuli should be explained in more detail.

We thank the reviewer for raising this point about the large stimulus arrays. As we said in our Discussion, it would seem that aggregation across a large stimulus region would be better suited to a downstream visual area with larger receptive fields, yet our setting of a strict deadline would put the emphasis back on earlier sensory representations. We now elaborate on this matter in the discussion, to say that although the small receptive fields and short, slow horizontal connections in V1 mean that the aggregation necessary for performing the task is unlikely to happen within V1 during the C1 timeframe, the aggregation would be readily achieved simply by convergence of the outputs of all relevant V1 neurons for a given stimulus array on the same decision process. In this sense, the design of our paradigm was such that the globally-measured C1 component on the scalp reflected the same aggregated evidence input as the summed V1 readout that we suppose would be entering the decision process.

We have also added further rationale in the Methods section on the practical benefits of the stimulus design, as the reviewer anticipates in their subsequent point, of yielding robust C1 signals. This concern was paramount in the design of this study because we expected the C1 difference metric that was of interest to be very small. We also needed a robust C1 to be measured in both the upper and lower visual field in as many individuals as possible and, in our experience, this is true less often when using smaller stimuli, even with a pre-mapping procedure.

It also helped to homogenize C1 topography across individuals and ensure that topographies from the upper and lower visual field had sufficient overlap that there were electrodes with strong loading from both topographies where the C1 difference as a function of which array was brighter would be maximal.

We have updated the methods section to provide these rationales while we describe the stimulus design.

(2) In a similar vein, while C1 has canonical topographical properties at the grand-average level, these may differ substantially depending on individual anatomy (which the authors did not assess). This means that task-relevant information will be represented to different degrees in individuals' single-trial data. My guess is that this confound was mitigated precisely by choosing relatively extended stimulus arrays. But given the authors' impressive track record on C1 mapping and modeling, I was surprised that the underlying rationale is only roughly outlined. For example, given the topographies shown and the electrode selection procedure employed, I assume that the differences between upper and lower targets are mainly driven by stimulus arms on the main diagonal. Did the authors run pilot experiments with more restricted stimulus arrays? I do not mean to imply that such additional information needs to be detailed in the main article, but it would be worth mentioning.

We thank the reviewer for their thoughtful consideration of this issue about individual variability in C1 retinotopy. Indeed, as the reviewer anticipated we expected the large stimulus coverage to mitigate this issue and we think that our response to the point above and the changes we made to the manuscript in response address this point also. Although we did not show this in the manuscript, we did in fact find that C1 topography was much more similar across individuals than it has been in previous C1 experiments we have carried out with smaller stimuli.

However, we acknowledge the reviewer’s point that the signal measured at a specific electrode likely has a variable loading strength from the various gratings in the stimulus array and that the gratings of maximal loading may indeed vary from subject to subject. Such inter-subject variability cannot confound the choice probability effects because the latter are measured within-subject. Nevertheless, it could be a source of noise. We believe the impact of this is unlikely to be substantial for the following reasons:

i) We designed the spatial spread of contrasts in such a way as to encourage participants to aggregate across the full array. In essence, to match the property of the C1 as an aggregate measure of V1 activity, we designed a task that involved aggregating across stimulus elements. Therefore, the decision weighting applied to any particular grating should be representative of the weighting applied to all gratings and, as such, the specific gratings that contribute most to the C1 signal for a particular participant should be relatively inconsequential.

ii) By avoiding the horizontal and vertical meridians we avoided the regions of space where the shifts in C1 topography are largest.

(3) Also, the stimulus arrangement disregards known differences in conduction velocity between the upper and lower visual fields. While no such differences are evident from the maximal-electrode averages shown in Figure 1B, it is difficult to assess this issue without single-stimulus VEPs and/or a dedicated latency analysis. The authors touch upon this issue when discussing potential pre-C1 signals emanating from the magnocellular pathway.

Indeed, there are important differences in V1 properties between the upper and lower visual fields, visual acuity being another example in addition to conduction velocity as the reviewer points out. However, these differences appeared to be quite minimal in this case (Figure 1B does in fact include a single-stimulus VEP – the “1-stim” entry in the legend). Perhaps this is also due to the large stimulus array which may include a range of conduction velocities within it and thereby blur overall differences between the upper and lower visual field. The variability of contrast within each array was also quite high (+/-20% from the midpoint), which would have further increased within-array conduction velocity variability and blurred differences between arrays.

Our staircasing procedure may have also helped in this regard to some extent as it included a bias parameter between the arrays to account for any behavioural response biases. Although the small contrast changes it usually incurred are likely much too small to change conduction velocities, it corrected for any effect on behaviour they may have.

(4) I suspect that most of these issues are at least partly related to a lack of clarity regarding levels of description: the authors often refer to 'information' contained in C1 or, apparently interchangeably, to 'visual representations' before, during, or following C1. However, if I understand correctly, the signal predicting (or predicted by) behavioral choice is much cruder than what an RSA-primed readership may expect, and also cruder than the other choice-predictive signals entered as control variables: namely, a univariate difference score on single-trial data integrated over a 10 ms window determined on the basis of grand-averaged data. I think it is worth clarifying and emphasizing the nature of this signal as the difference of aggregate contrast responses that *can* only be read out at higher levels of the visual system due to the limited extent of horizontal connectivity in V1. I do not think that this diminishes the importance of the findings - if anything, it makes them more remarkable.

This is true that a univariate measure may stick out in a field increasingly favouring multivariate analyses with the spread of machine learning, and so we have added a short qualifier in the methods section where we describe the C1 measurement to explicitly state that it is a scalar variable. What we have done in using this univariate measure is leverage the rich prior knowledge about V1 anatomy and neurophysiology, rather than trust in data-driven classifiers; interestingly, we found that such a classifier trained on all electrodes discriminates choices less well than our informed univariate measure during the C1 time-frame.

We also thank the reviewer for raising an interesting point about the nature of aggregation and readout in the context of our stimulus. We agree that it is not feasible that V1 activity would be aggregated locally in V1 across such large regions of space prior to being readout within the C1 time period. As we say above, the aggregation may instead be carried out through convergent transmission of the parallel, spatially-local V1 information to the decision process.

(5) Arguably even more remarkable is the finding that C1 amplitudes themselves appear to be influenced by choice history. The authors address this issue in the Discussion; however, I'm afraid I could not follow their argument regarding preparatory (and differential?) weighting of read-outs across the visual hierarchy. I believe this point is worth developing further, as it bears on the issue of whether C1 modulations are present and ecologically relevant when looking (before and) beyond stimulus-locked averages.

We thank the reviewer for their positive appraisal of this additional finding, which we also found remarkable. We agree that our description of our interpretation was too brief and lacked clarity. We have reworded it and expressed it in terms of the speed accuracy trade-off, with the new explanation given below. However, it is important to remember that this account is speculative and serves only to explain the response-time contingency of the bias. That the bias was present and constitutes a modulation of the C1 does not rest on this argument:

[…] “to explain the RT contingency for the C1 bias, we speculate that the speed-accuracy trade-off could fluctuate from trial to trial and that the corresponding decision bound fluctuations (Heitz and Schall 2012) could be implemented by pre-determining decision weights across visual areas. For example, to achieve faster decisions, the sensory evidence requirement could be reduced by placing greater emphasis on initial afferent V1 evidence. In such a case, the RT contingency of the above choice history bias could be explained if the C1 bias is exerted in proportion with the planned emphasis of C1 evidence for the upcoming decision.”

**Recommendations to the Authors:**

**Reviewer #2 (Recommendations for the authors):**
(1) As someone whose first language is not English, I am somewhat hesitant to bring this up, but I found the use of 'readout' as both noun and verb somewhat confusing. I thought read-out was defined as 'that which is read out'.

We agree that this dual use of the word readout may cause confusion. To avoid this, we have edited the manuscript to replace verbal forms of the word “readout” with “read out”.

(2) I found it difficult to follow the reasoning for why intermediate RTs should be the ones most affected by C1-related information. Perhaps this could be described in more detail for the uninitiated reader.

We appreciate that our reasoning for why intermediate RTs should be the ones most affected by C1-related information was difficult to follow. We have now added a simulation to showcase this rationale more clearly - see response to reviewer 1, and new figure supplement to figure 1.

(3) It would be interesting to compare the effect sizes observed here to those seen in single-cell studies and to discuss this comparison with regard to differences in the nature of EEG signals and single-cell firing rates.

While we agree that such a comparison would be interesting if feasible, it would have to be for the same task settings, which have not been used in a single-cell study, and the very different nature and extent of noise between the two recording modalities would make such a comparison difficult to interpret, e.g. background noise in EEG from ongoing processes unrelated to the task.

(4) Figure 1: It may be worth mentioning in the legend that only parts of the peripheral stimulus grid are shown for better visibility, as the Methods speak of 9 x 9 grids. Also, in panel B, it should be mentioned that waveshapes are calculated using individually selected maximal-difference electrodes.

We thank the reviewer for spotting these. We have updated the caption for this figure to reflect these two observations.

(5) Figure 4: The different shades of green may be difficult to distinguish when printed.

Although this may be true, we chose shades of green that differ in luminance so they should still be distinguishable. Different colours may in fact be less distinguishable if they had the same luminance and the print was black-and-white. We chose different shades of the same colour to reflect the fact that we were plotting the same signals at different difficulty levels. In our opinion, this takes precedence since eLife is an online journal so the majority of readers will likely read it digitally.

(6) Methods/Task: While the ITI of 780 ms is substantial, I was wondering why the authors decided against jittering this interval? It would be helpful to briefly discuss whether contrast adaptation for slow periodic stimulation may have affected the findings.

We opted against jittering the ITI to avoid an additional source of inter-trial variability. While this may allow for adaptation effects of this source, this would be approximately constant across trials and therefore less of a concern for our design. We have added text to the methods section to state this rationale.

(7) Methods/Stimuli: The authors convincingly argue that focusing on single arms of the stimuli is an unlikely strategy, but did they ask for participants' strategies during debriefing?

We are glad that the reviewer found our argument about whether or not participants may have focused on a single arm of the stimuli convincing. We did not ask participants about their strategies but even with such a debriefing, there would still remain a possibility that a participant may have used that strategy but were unaware that they were doing so. In any case, if participants were doing this it would have dampened the strength of our choice probability result.

(8) Methods/Procedure, Difficulty Titration: Why did the authors opt for manually adapting the difficulty level in a separate session rather than constantly and automatically titrating difficulty?

We did this because calculating choice probability requires a comparison of trials with different choice outcomes but the same stimulus so continuously staircasing difficulty level during the experiment would have created a confound. Although this could have been corrected for in our regression, this would have entailed greater noise that we could avoid by staircasing in advance.